# Matrix Completion with Noisy Side Information

**Kai-Yang Chiang**[*]    **Cho-Jui Hsieh** [†]    **Inderjit S. Dhillon** [*]

[*] University of Texas at Austin        [†] University of California at Davis

[*] {kychiang,inderjit}@cs.utexas.edu

[†] chohsieh@ucdavis.edu

## Abstract

We study the matrix completion problem with side information. Side information has been considered in several matrix completion applications, and has been empirically shown to be useful in many cases. Recently, researchers studied the effect of side information for matrix completion from a theoretical viewpoint, showing that sample complexity can be significantly reduced given completely clean features. However, since in reality most given features are noisy or only weakly informative, the development of a model to handle a *general* feature set, and investigation of how much noisy features can help matrix recovery, remains an important issue. In this paper, we propose a novel model that balances between features and observations simultaneously in order to leverage feature information yet be robust to feature noise. Moreover, we study the effect of general features in theory and show that by using our model, the sample complexity can be lower than matrix completion as long as features are sufficiently informative. This result provides a theoretical insight into the usefulness of general side information. Finally, we consider synthetic data and two applications — relationship prediction and semi-supervised clustering — and show that our model outperforms other methods for matrix completion that use features both in theory and practice.

## 1   Introduction

Low rank matrix completion is an important topic in machine learning and has been successfully applied to many practical applications [22, 12, 11]. One promising direction in this area is to exploit the *side information*, or *features*, to help matrix completion tasks. For example, in the famous Netflix problem, besides rating history, profile of users and/or genre of movies might also be given, and one could possibly leverage such side information for better prediction. Observing the fact that such additional features are usually available in real applications, how to better incorporate features into matrix completion becomes an important problem with both theoretical and practical aspects.

Several approaches have been proposed for matrix completion with side information, and most of them empirically show that features are useful for certain applications [1, 28, 9, 29, 33]. However, there is surprisingly little analysis on the effect of features for general matrix completion. More recently, Jain and Dhillon [18] and Xu et al. [35] provided non-trivial guarantees on matrix completion with side information. They showed that if "perfect" features are given, under certain conditions, one can substantially reduce the sample complexity by solving a feature-embedded objective. This result suggests that completely informative features are extremely powerful for matrix completion, and the algorithm has been successfully applied in many applications [29, 37]. However, this model is still quite restrictive since if features are not perfect, it fails to guarantee recoverability and could even suffer poor performance in practice. A more general model with recovery analysis to handle noisy features is thus desired.

In this paper, we study the matrix completion problem with *general* side information. We propose a dirty statistical model which balances between feature and observation information simultaneously to complete a matrix. As a result, our model can leverage feature information, yet is robust to noisy features. Furthermore, we provide a theoretical foundation to show the effectiveness of our model. We formally quantify the quality of features and show that the sample complexity of our model

depends on feature quality. Two noticeable results could thus be inferred: first, unlike [18, 35], given any feature set, our model is guaranteed to achieve recovery with at most $O(n^{3/2})$ samples in *distribution-free* manner, where $n$ is the dimensionality of the matrix. Second, if features are reasonably good, we can improve the sample complexity to $o(n^{3/2})$. We emphasize that since $\Omega(n^{3/2})$ is the lower bound of sample complexity for distribution-free, trace-norm regularized matrix completion [32], our result suggests that even noisy features could asymptotically reduce the number of observations needed in matrix completion. In addition, we empirically show that our model outperforms other completion methods on synthetic data as well as in two applications: relationship prediction and semi-supervised clustering. Our contribution can be summarized as follows:

- We propose a dirty statistical model for matrix completion with *general* side information where the matrix is learned by balancing features and pure observations simultaneously.
- We quantify the effectiveness of features in matrix completion problem.
- We show that our model is guaranteed to recover the matrix with any feature set, and moreover, the sample complexity can be lower than standard matrix completion given informative features.

The paper is organized as follows. Section 2 states some related research. In Section 3, we introduce our proposed model for matrix completion with general side information. We theoretically analyze the effectiveness of features in our model in Section 4, and show experimental results in Section 5.

## 2  Related Work

Matrix completion has been widely applied to many machine learning tasks, such as recommender systems [22], social network analysis [12] and clustering [11]. Several theoretical foundations have also been established. One remarkable milestone is the strong guarantee provided by Candès et al. [7, 5], who proves that $O(n\,\mathrm{polylog}\,n)$ observations are sufficient for exact recovery provided entries are uniformly sampled at random. Several work also studies recovery under non-uniform distributional assumptions [30, 10], distribution-free setting [32], and noisy observations [21, 4].

Several works also consider side information in matrix completion [1, 28, 9, 29, 33]. Although most of them found that features are helpful for certain applications [28, 33] and cold-start setting [29] from their experimental supports, their proposed methods focus on the non-convex matrix factorization formulation without any theoretical guarantees. Compared to them, our model mainly focuses on a convex trace-norm regularized objective and on theoretical insight on the effect of features. On the other hand, Jain and Dhillon [18] (also see [38]) studied an inductive matrix completion objective to incorporate side information, and followup work [35] also considers a similar formulation with trace norm regularized objective. Both of them show that recovery guarantees could be attained with lower sample complexity when features are perfect. However, if features are imperfect, such models cannot recover the underlying matrix and could suffer poor performance in practice. We will have a detailed discussion on inductive matrix completion model in Section 3.

Our proposed model is also related to the family of dirty statistical models [36], where the model parameter is expressed as the sum of a number of parameter components, each of which has its own structure. Dirty statistical models have been proposed mostly for robust matrix completion, graphical model estimation, and multi-task learning to decompose the sparse component (noise) and low-rank component (model parameters) [6, 8, 19]. Our proposed algorithm is completely different. We aim to decompose the model into two parts: the part that can be described by side information and the part that has to be recovered purely by observations.

## 3  A Dirty Statistical Model for Matrix Completion with Features

Let $R \in \mathbb{R}^{n_1 \times n_2}$ be the underlying rank-$k$ matrix that aims to be recovered, where $k \ll \min(n_1, n_2)$ so that $R$ is low-rank. Let $\Omega$ be the set of observed entries sampled from $R$ with cardinality $|\Omega| = m$. Furthermore, let $X \in \mathbb{R}^{n_1 \times d_1}$ and $Y \in \mathbb{R}^{n_2 \times d_2}$ be the feature set, where each row $\mathbf{x}_i$ (or $\mathbf{y}_i$) denotes the feature of the $i$-th row (or column) entity of $R$. Both $d_1, d_2 \leq \min(n_1, n_2)$ but can be either smaller or larger than $k$. Thus, given a set of observations $\Omega$ and the feature set $X$ and $Y$ as side information, the goal is to recover the underlying low rank matrix $R$.

To begin with, consider an ideal case where the given features are "perfect" in the following sense:

$$\mathrm{col}(R) \subseteq \mathrm{col}(X) \quad \text{and} \quad \mathrm{row}(R) \subseteq \mathrm{col}(Y). \tag{1}$$

Such a feature set can be thought as perfect since it fully describes the true latent feature space of $R$. Then, instead of recovering the low rank matrix $R$ directly, one can recover a smaller matrix

$M \in \mathbb{R}^{d_1 \times d_2}$ such that $R = XMY^T$. The resulting formulation, called inductive matrix completion (or IMC in brief) [18], is shown to be both theoretically preferred [18, 35] and useful in real applications [37, 29]. Details of this model can be found in [18, 35].

However, in practice, most given features $X$ and $Y$ will not be perfect. In fact, they could be quite noisy or only weakly correlated to the latent feature space of $R$. Though in some cases applying IMC with imperfect $X, Y$ might still yield decent performance, in many other cases, the performance drastically drops when features become noisy. This weakness of IMC can also be empirically seen in Section 5. Therefore, a more robust model is desired to better handle noisy features.

We now introduce a dirty statistical model for matrix completion with (possibly noisy) features. The core concept of our model is to learn the underlying matrix by balancing feature information and observations. Specifically, we propose to learn $R$ jointly from two parts, one is the low rank estimate from feature space $XMY^T$, and the other part $N$ is the part outside the feature space. Thus, $N$ can be used to capture the information that noisy features fail to describe, which is then estimated by pure observations. Naturally, both $XMY^T$ and $N$ are preferred to be low rank since they are aggregated to estimate a low rank matrix $R$. This further leads a preference on $M$ to be low rank as well, since one could expect only a small subspace of $X$ and a subspace of $Y$ are jointly effective to form the low rank space $XMY^T$. Putting all of above together, we consider to solve the following problem:

$$\min_{M,N} \sum_{(i,j) \in \Omega} \ell((XMY^T + N)_{ij}, R_{ij}) + \lambda_M \|M\|_* + \lambda_N \|N\|_*, \tag{2}$$

where $M$ and $N$ are regularized with trace norm because of the low rank prior. The underlying matrix $R$ can thus be estimated by $XM^*Y^T + N^*$. We refer our model as DirtyIMC for convenience.

To solve the convex problem (2), we propose an alternative minimization scheme to solve $N$ and $M$ iteratively. Our algorithm is stated in details in Appendix A. One remark of this algorithm is that it is guaranteed to converge to a global optimal, since the problem is jointly convex with $M$ and $N$.

The parameters $\lambda_M$ and $\lambda_N$ are crucial for controlling the importance between features and residual. When $\lambda_M = \infty$, $M$ will be enforced to 0, so features are disregarded and (2) becomes a standard matrix completion objective. Another special case is $\lambda_N = \infty$, in which $N$ will be enforced to 0 and the objective becomes IMC. Intuitively, with an appropriate ratio $\lambda_M/\lambda_N$, the proposed model can incorporate useful part of features, yet be robust to noisy part by compensating from pure observations. Some natural questions arise from here: How to quantify the quality of features? What is the right $\lambda_M$ and $\lambda_N$ given a feature set? And beyond intuition, how much can we benefit from features using our model *in theory*? We will formally answer these questions in Section 4.

## 4 Theoretical Analysis

Now we analyze the usefulness of features in our model under a theoretical perspective. We first quantify the quality of features and show that with reasonably good features, our model achieves recovery with lower sample complexity. Finally, we compare our results to matrix completion and IMC. Due to space limitations, detailed proofs of Theorems and Lemmas are left in Appendix B.

### 4.1 Preliminaries

Recall that our goal is to recover a rank-$k$ matrix $R$ given observed entry set $\Omega$, feature set $X$ and $Y$ described in Section 3. To recover the matrix with our model (Equation (2)), it is equivalent to solve the hard-constraint problem:

$$\min_{M,N} \sum_{(i,j) \in \Omega} \ell((XMY^T + N)_{ij}, R_{ij}), \quad \text{subject to } \|M\|_* \leq \mathcal{M}, \|N\|_* \leq \mathcal{N}. \tag{3}$$

For simplicity, we will consider $d = \max(d_1, d_2) = O(1)$ so that feature dimensions do not grow as a function of $n$. We assume each entry $(i,j) \in \Omega$ is sampled i.i.d. under an unknown distribution with index set $\{(i_\alpha, j_\alpha)\}_{\alpha=1}^m$. Also, each entry of $R$ is assumed to be upper bounded, i.e. $\max_{ij} |R_{ij}| \leq \mathcal{R}$ (so that trace norm of $R$ is in $O(\sqrt{n_1 n_2})$). Such circumstance is consistent with real scenarios like the Netflix problem where users can rate movies with scale from 1 to 5. For convenience, let $\theta = (M, N)$ be any feasible solution, and $\Theta = \{(M, N) \mid \|M\|_* \leq \mathcal{M}, \|N\|_* \leq \mathcal{N}\}$ be the feasible solution set. Also, let $f_\theta(i, j) = \mathbf{x}_i^T M \mathbf{y}_j + N_{ij}$ be the estimation function for $R_{ij}$ parameterized by $\theta$, and $F_\Theta = \{f_\theta \mid \theta \in \Theta\}$ be the set of feasible functions. We are interested in the following two "$\ell$-risk" quantities:

- Expected $\ell$-risk: $R_\ell(f) = \mathbb{E}_{(i,j)} \big[ \ell(f(i, j), R_{ij}) \big]$.

- Empirical $\ell$-risk: $\hat{R}_\ell(f) = \frac{1}{m}\sum_{(i,j)\in\Omega}\ell(f(i,j), R_{ij})$.

Thus, our model is to solve for $\theta^*$ that parameterizes $f^* = \arg\min_{f\in F_\Theta}\hat{R}_\ell(f)$, and it is sufficient to show that recovery can be attained if $R_\ell(f^*)$ approaches to zero with large enough $n$ and $m$.

## 4.2 Measuring the Quality of Features

We now link the quality of features to Rademacher complexity, a learning theoretic tool to measure the complexity of a function class. We will show that quality features result in a lower model complexity and thus a smaller error bound. Under such a viewpoint, the upper bound of Rademacher complexity could be used for measuring the quality of features.

To begin with, we apply the following Lemma to bound the expected $\ell$-risk.

**Lemma 1** (Bound on Expected $\ell$-risk [2]). *Let $\ell$ be a loss function with Lipschitz constant $L_\ell$ bounded by $\mathcal{B}$ with respect to its first argument, and $\delta$ be a constant where $0 < \delta < 1$. Let $\mathfrak{R}(F_\Theta)$ be the Rademacher complexity of the function class $F_\Theta$ (w.r.t. $\Omega$ and associated with $\ell$) defined as:*

$$\mathfrak{R}(F_\Theta) = \mathbb{E}_\sigma\Big[\sup_{f\in F_\Theta}\frac{1}{m}\sum_{\alpha=1}^{m}\sigma_\alpha\ell(f(i_\alpha, j_\alpha), R_{i_\alpha j_\alpha})\Big], \tag{4}$$

*where each $\sigma_\alpha$ takes values $\{\pm 1\}$ with equal probability. Then with probability at least $1 - \delta$, for all $f \in F_\Theta$ we have:*

$$R_\ell(f) \le \hat{R}_\ell(f) + 2\mathbb{E}_\Omega\big[\mathfrak{R}(F_\Theta)\big] + \mathcal{B}\sqrt{\frac{\log\frac{1}{\delta}}{2m}}.$$

Apparently, to guarantee a small enough $R_\ell$, both $\hat{R}_\ell$ and model complexity $\mathbb{E}_\Omega\big[\mathfrak{R}(F_\Theta)\big]$ have to be bounded. The next key lemma shows that, the model complexity term $\mathbb{E}_\Omega\big[\mathfrak{R}(F_\Theta)\big]$ is related to the feature quality in matrix completion context.

Before diving into the details, we first provide an intuition on the meaning of "good" features. Consider any imperfect feature set which violates (1). One can imagine such feature set is perturbed by some misleading noise which is not correlated to the true latent features. However, features should still be effective if such noise does not weaken the true latent feature information too much. Thus, if a large portion of true latent features lies on the informative part of the feature spaces $X$ and $Y$, they should still be somewhat informative and helpful for recovering the matrix $R$.

More formally, the model complexity can be bounded in terms of $\mathcal{M}$ and $\mathcal{N}$ by the following lemma:

**Lemma 2.** *Let $\mathcal{X} = \max_i\|\mathbf{x}_i\|_2$, $\mathcal{Y} = \max_i\|\mathbf{y}_i\|_2$ and $n = \max(n_1, n_2)$. Then the model complexity of function class $F_\Theta$ is upper bounded by:*

$$\mathbb{E}_\Omega\big[\mathfrak{R}(F_\Theta)\big] \le 2L_\ell\mathcal{M}\mathcal{X}\mathcal{Y}\sqrt{\frac{\log 2d}{m}} + \min\Big\{2L_\ell\mathcal{N}\sqrt{\frac{\log 2n}{m}}, \sqrt{9CL_\ell\mathcal{B}\frac{\mathcal{N}(\sqrt{n_1}+\sqrt{n_2})}{m}}\Big\}.$$

Then, by Lemma 1 and 2, one could carefully construct a feasible solution set (by setting $\mathcal{M}$ and $\mathcal{N}$) such that both $\hat{R}_\ell(f^*)$ and $\mathbb{E}_\Omega\big[\mathfrak{R}(F_\Theta)\big]$ are controlled to be reasonably small. We now suggest a witness pair of $\mathcal{M}$ and $\mathcal{N}$ constructed as follows. Let $\gamma$ be defined as:

$$\gamma = \min\Big(\frac{\min_i\|\mathbf{x}_i\|}{\mathcal{X}}, \frac{\min_i\|\mathbf{y}_i\|}{\mathcal{Y}}\Big).$$

Let $\mathcal{T}_\mu(\cdot) : \mathbb{R}^+ \to \mathbb{R}^+$ be the thresholding operator where $\mathcal{T}_\mu(x) = x$ if $x \ge \mu$ and $\mathcal{T}_\mu(x) = 0$ otherwise. In addition, let $X = \sum_{i=1}^{d_1}\sigma_i\mathbf{u}_i\mathbf{v}_i^T$ be the reduced SVD of $X$, and define $X_\mu = \sum_{i=1}^{d_1}\sigma_1\mathcal{T}_\mu(\sigma_i/\sigma_1)\mathbf{u}_i\mathbf{v}_i^T$ to be the "$\mu$-informative" part of $X$. The $\nu$-informative part of $Y$, denoted as $Y_\nu$, can also be defined similarly. Now consider setting $\mathcal{M} = \|\hat{M}\|_*$ and $\mathcal{N} = \|R - X_\mu\hat{M}Y_\nu^T\|_*$, where

$$\hat{M} = \arg\min_M\|X_\mu M Y_\nu^T - R\|_F^2 = (X_\mu^T X_\mu)^{-1}X_\mu^T R Y_\nu(Y_\nu^T Y_\nu)^{-1}$$

is the optimal solution for approximating $R$ under the informative feature space $X_\mu$ and $Y_\nu$. Then the following lemma shows that the trace norm of $\hat{M}$ will not grow as $n$ increases.

**Lemma 3.** *Fix $\mu, \nu \in (0, 1]$, and let $\hat{d} = \min(rank(X_\mu), rank(Y_\nu))$. Then with some universal constant $C'$:*

$$\|\hat{M}\|_* \le \frac{\hat{d}}{C'\mu^2\nu^2\gamma^2\mathcal{X}\mathcal{Y}}.$$

Moreover, by combining Lemma 1 - 3, we can upper bound $R_\ell(f^*)$ of DirtyIMC as follows:

**Theorem 1.** *Consider problem (3) with $\mathcal{M} = \|\hat{M}\|_*$ and $\mathcal{N} = \|R - X_\mu \hat{M} Y_\nu^T\|_*$. Then with probability at least $1 - \delta$, the expected $\ell$-risk of an optimal solution $(N^*, M^*)$ will be bounded by:*

$$R_\ell(f^*) \leq \min \left\{ 4L_\ell \mathcal{N} \sqrt{\frac{\log 2n}{m}}, \sqrt{36 C L_\ell \mathcal{B} \frac{\mathcal{N}(\sqrt{n_1} + \sqrt{n_2})}{m}} \right\} + \frac{4L_\ell \hat{d}}{C' \mu^2 \nu^2 \gamma^2} \sqrt{\frac{\log 2d}{m}} + \mathcal{B} \sqrt{\frac{\log \frac{1}{\delta}}{2m}}.$$

### 4.3 Sample Complexity Analysis

From Theorem 1, we can derive the following sample complexity guarantee of our model. For simplicity, we assume $k = O(1)$ so it will not grow as $n$ increases in the following discussion.

**Corollary 1.** *Suppose we aim to "$\epsilon$-recover" $R$ where $\mathbb{E}_{(i,j)}\left[\ell(N_{ij} + XMY_{ij}^T, R_{ij})\right] < \epsilon$ given an arbitrarily small $\epsilon$. Then for DirtyIMC model, $O(\min(\mathcal{N}\sqrt{n}, \mathcal{N}^2 \log n)/\epsilon^2)$ observations are sufficient for $\epsilon$-recovery provided a sufficiently large $n$.*

Corollary 1 suggests that the sample complexity of our model only depends on the trace norm of residual $\mathcal{N}$. This matches the intuition of good features stated in Section 4.2 because $X\hat{M}Y^T$ will cover most part of $R$ if features are good, and as a result, $\mathcal{N}$ will be small and one can enjoy small sample complexity by exploiting quality features.

We also compare our sample complexity result with other models. First, suppose features are perfect (so that $\mathcal{N} = O(1)$), our result suggests that only $O(\log n)$ samples are required for recovery. This matches the result of [35], in which the authors show that given perfect features, $O(\log n)$ observations are enough for exact recovery by solving the IMC objective. However, IMC does not guarantee recovery when features are not perfect, while our result shows that recovery is still attainable by DirtyIMC with $O(\min(\mathcal{N}\sqrt{n}, \mathcal{N}^2 \log n)/\epsilon^2)$ samples. We will also empirically justify this result in Section 5.

On the other hand, for standard matrix completion (i.e. no features are considered), the most well-known guarantee is that under certain conditions, one can achieve $O(n \operatorname{poly} \log n)$ sample complexity for both $\epsilon$-recovery [34] and exact recovery [5]. However, these bounds only hold with distributional assumptions on observed entries. For sample complexity without any distributional assumptions, Shamir et al. [32] recently showed that $O(n^{3/2})$ entries are sufficient for $\epsilon$-recovery, and this bound is tight if no further distribution of observed entries is assumed. Compared to those results, our analysis also requires no assumptions on distribution of observed entries, and our sample complexity yields $O(n^{3/2})$ as well in the worst case, by the fact that $\mathcal{N} \leq \|R\|_* = O(n)$. Notice that it is reasonable to meet the lower bound $\Omega(n^{3/2})$ even given features, since in an extreme case, $X, Y$ could be random matrices and have no correlation to $R$, and thus the given information is as same as that in standard matrix completion.

However, in many applications, features will be far from random, and our result provides a theoretical insight to show that features can be useful even if they are imperfect. Indeed, as long as features are informative enough such that $\mathcal{N} = o(n)$, our sample complexity will be asymptotically lower than $O(n^{3/2})$. Here we provide two concrete instances for such a scenario. In the first scenario, we consider the rank-$k$ matrix $R$ to be generated from random orthogonal model [5] as follows:

**Theorem 2.** *Let $R \in \mathbb{R}^{n \times n}$ be generated from random orthogonal model, where $U = \{\mathbf{u}_i\}_{i=1}^k$, $V = \{\mathbf{v}_i\}_{i=1}^k$ are random orthogonal bases, and $\sigma_1 \ldots \sigma_k$ are singular values with arbitrary magnitude. Let $\sigma_t$ be the largest singular value such that $\lim_{n \to \infty} \sigma_t/\sqrt{n} = 0$. Then, given the noisy features $X, Y$ where $X_{:i} = \mathbf{u}_i$ (and $Y_{:i} = \mathbf{v}_i$) if $i < t$ and $X_{:i}$ (and $V_{:i}$) be any basis orthogonal to $U$ (and $V$) if $i \geq t$, $o(n)$ samples are sufficient for DirtyIMC to achieve $\epsilon$-recovery.*

Theorem 2 suggests that, under random orthogonal model, if features are not too noisy in the sense that noise only corrupts the true subspace associated with smaller singular values, we can approximately recover $R$ with only $o(n)$ observations. An empirical justification for this result is presented in Appendix C. Another scenario is to consider $R$ to be the product of two rank-$k$ Gaussian matrices:

**Theorem 3.** *Let $R = UV^T$ be a rank-$k$ matrix, where $U, V \in \mathbb{R}^{n \times k}$ are true latent row/column features with each $U_{ij}, V_{ij} \sim \mathcal{N}(0, \sigma^2)$ i.i.d. Suppose now we are given a feature set $X, Y$ where $g(n)$ row items and $h(n)$ column items have corrupted features. Moreover, each corrupted row/column item has perturbed feature $\mathbf{x}_i = \mathbf{u}_i + \Delta \mathbf{u}_i$ and $\mathbf{y}_i = \mathbf{v}_i + \Delta \mathbf{v}_i$, where $\|\Delta \mathbf{u}\|_\infty \leq \xi_1$ and*

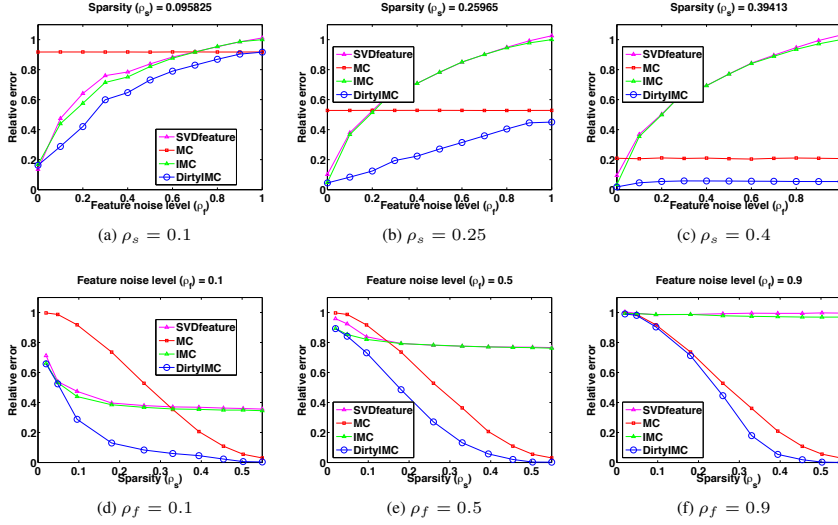

Figure 1: Performance of various methods for matrix completion under different sparsity and feature quality. Compared to other feature-based completion methods, the top figures show that DirtyIMC is less sensitive to noisy features with each $\rho_s$, and the bottom figures show that error of DirtyIMC always decreases to 0 with more observations given any feature quality.

$\|\Delta \mathbf{v}\|_\infty \leq \xi_2$ *with some constants $\xi_1$ and $\xi_2$. Then for DirtyIMC model* (3)*, with high probability,* $O\big(\max(\sqrt{g(n)}, \sqrt{h(n)})n \log n\big)$ *observations are sufficient for $\epsilon$-recovery.*

Theorem 3 suggests that, if features have good quality in the sense that items with corrupted features are not too many, for example $g(n), h(n) = O(\log n)$, then sample complexity of DirtyIMC can be $O(n \log n \sqrt{\log n}) = o(n^{3/2})$ as well. Thus, both Theorem 2 and 3 provide concrete examples showing that given imperfect yet informative features, the sample complexity of our model can be asymptotically lower than the lower bound of pure matrix completion (which is $\Omega(n^{3/2})$).

## 5 Experimental Results

In this section, we show the effectiveness of the DirtyIMC model (2) for matrix completion with features on both synthetic datasets and real-world applications. For synthetic datasets, we show that DirtyIMC model better recovers low rank matrices under various quality of features. For real applications, we consider relationship prediction and semi-supervised clustering, where the current state-of-the-art methods are based on matrix completion and IMC respectively. We show that by applying DirtyIMC model to these two problems, we can further improve performance by making better use of features.

### 5.1 Synthetic Experiments

We consider matrix recovery with features on synthetic data generated as follows. We create a low rank matrix $R = UV^T$, as the true latent row/column space $U, V \in \mathbb{R}^{200 \times 20}$, $U_{ij}, V_{ij} \sim \mathcal{N}(0, 1/20)$. We then randomly sample $\rho_s$ percent of entries $\Omega$ from $R$ as observations, and construct a perfect feature set $X^*, Y^* \in \mathbb{R}^{200 \times 40}$ which satisfies (1). To examine performance under different quality of features, we generate features $X, Y$ with a noise parameter $\rho_f$, where $X$ and $Y$ will be derived by replacing $\rho_f$ percent of bases of $X^*$ (and $Y^*$) with bases orthogonal to $X^*$ (and $Y^*$). We then consider recovering the underlying matrix $R$ given $X, Y$ and a subset $\Omega$ of $R$.

We compare our DirtyIMC model (2) with standard trace-norm regularized matrix completion (MC) and two other feature-based completion methods: IMC [18] and SVDfeature [9]. The standard relative error $\|\hat{R} - R\|_F / \|R\|_F$ is used to evaluate a recovered matrix $\hat{R}$. For each method, we select parameters from the set $\{10^\alpha\}_{\alpha=-3}^2$ and report the one with the best recovery. All results are averaged over 5 random trials.

Figure 1 shows the recovery of each method under each sparsity level $\rho_s = 0.1, 0.25, 0.4$, and each feature noise level $\rho_f = 0.1, 0.5$ and $0.9$. We first observe that in the top figures, IMC and

| Method | DirtyIMC | MF-ALS [16] | IMC [18] | HOC-3 | HOC-5 [12] |
|---|---|---|---|---|---|
| Accuracy | **0.9474**±0.0009 | 0.9412±0.0011 | 0.9139±0.0016 | 0.9242±0.0010 | 0.9297±0.0011 |
| AUC | **0.9506** | 0.9020 | 0.9109 | 0.9432 | 0.9480 |

Table 1: Relationship prediction on Epinions. Compared with other approaches, DirtyIMC model gives the best performance in terms of both accuracy and AUC.

SVDfeature perform similarly under different $\rho_s$. This suggests that with sufficient observations, performance of IMC and SVDfeature mainly depend on feature quality and will not be affected much by the number of observations. As a result, given good features (1d), they achieve smaller error compared to MC with few observations, but as features become noisy (1e-1f), they suffer poor performance by trying to learn the underlying matrix under biased feature spaces. Another interesting finding is that when good features are given (1d), IMC (and SVDfeature) still fails to achieve 0 relative error as the number of observations increases, which reconfirms that IMC cannot guarantee recoverability when features are not perfect. On the other hand, we see that performance of DirtyIMC can be improved by both better features or more observations. In particular, it makes use of informative features to achieve lower error compared to MC and is also less sensitive to noisy features compared to IMC and SVDfeature. Some finer recovery results on $\rho_s$ and $\rho_f$ can be found in Appendix C.

## 5.2 Real-world Applications

**Relationship Prediction in Signed Networks.** As the first application, we consider relationship prediction problem in an online review website Epinions [26], where people can write reviews and trust or distrust others based on their reviews. Such social network can be modeled as a signed network where trust/distrust are modeled as positive/negative edges between entities [24], and the problem is to predict unknown relationship between any two users given the network. A state-of-the-art approach is the low rank model [16, 12] where one can first conduct matrix completion on adjacency matrix and then use the sign of completed matrix for relationship prediction. Therefore, if features of users are available, we can also consider low rank model by using our model for matrix completion step. This approach can be regarded as an improvement over [16] by incorporating feature information.

In this dataset, there are about $n = 105K$ users and $m = 807K$ observed relationship pairs where 15% relationships are distrust. In addition to who-trust-to-whom information, we also have user feature matrix $Z \in \mathbb{R}^{n \times 41}$ where for each user a 41-dimensional feature is collected based on the user's review history, such as number of positive/negative reviews the user gave/received. We then consider the low-rank model in [16] where matrix completion is conducted by DirtyIMC with non-convex relaxation (5) (DirtyIMC), IMC [18] (IMC), and matrix factorization proposed in [16] (MF-ALS), along with another two prediction methods, HOC-3 and HOC-5 [12]. Note that both row and column entities are users so $X = Y = Z$ is set for both DirtyIMC and IMC model.

We conduct the experiment using 10-fold cross validation on observed edges, where the parameters are chosen from the set $\sqcup_{\alpha=-3}^{2}\{10^\alpha, 5 \times 10^\alpha\}$. The averaged accuracy and AUC of each method are reported in Table 1. We first observe that IMC performs worse than MF-ALS even though IMC takes features into account. This is because features are only weakly related to relationship matrix, and as a result, IMC is misled by such noisy features. On the other hand, DirtyIMC performs the best among all prediction methods. In particular, it performs slightly better than MF-ALS in terms of accuracy, and much better in terms of AUC. This shows DirtyIMC can still exploit weakly informative features without being trapped by noisy features.

**Semi-supervised Clustering.** We now consider semi-supervised clustering problem as another application. Given $n$ items, the item feature matrix $Z \in \mathbb{R}^{n \times d}$, and $m$ pairwise constraints specifying whether item $i$ and $j$ are similar or dissimilar, the goal is to find a clustering of items such that most similar items are within the same cluster.

We notice that the problem can indeed be solved by matrix completion. Consider $S \in \mathbb{R}^{n \times n}$ to be the signed similarity matrix defined as $S_{ij} = 1$ (or $-1$) if item $i$ and $j$ are similar (or dissimilar), and 0 if similarity is unknown. Then solving semi-supervised clustering becomes equivalent to finding a clustering of the symmetric signed graph $S$, where the goal is to cluster nodes so that most edges within the same group are positive and most edges between groups are negative [12]. As a result, a matrix completion approach [12] can be applied to solve the signed graph clustering problem on $S$.

Apparently, the above solution is not optimal for semi-supervised clustering as it disregards features. Many semi-supervised clustering algorithms are thus proposed by taking both item features

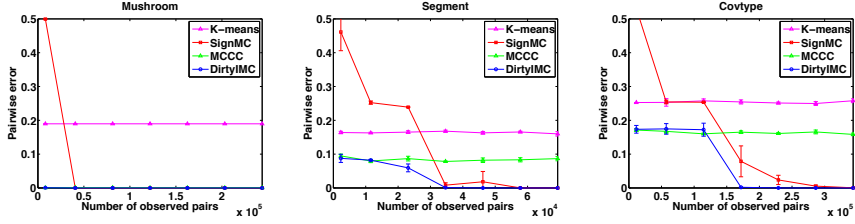

Figure 2: Semi-supervised clustering on real-world datasets. For Mushroom dataset where features are almost ideal, both MCCC and DirtyIMC achieve 0 error rate. For Segment and Covtype where features are more noisy, our model outperforms MCCC as its error decreases given more constraints.

| | number of items $n$ | feature dimension $d$ | number of clusters $k$ |
|---|---|---|---|
| Mushrooms | 8124 | 112 | 2 |
| Segment | 2319 | 19 | 7 |
| Covtype | 11455 | 54 | 7 |

Table 2: Statistics of semi-supervised clustering datasets.

and constraints into consideration [13, 25, 37]. The current state-of-the-art method is the MCCC algorithm [37], which essentially solves semi-supervised clustering with IMC objective. In [37], the authors show that by running $k$-means on the top-$k$ eigenvectors of the completed matrix $ZMZ^T$, MCCC outperforms other state-of-the-art algorithms [37].

We now consider solving semi-supervised clustering with our DirtyIMC model. Our algorithm, summarized in Algorithm 2 in Appendix D, first completes the pairwise matrix with DirtyIMC objective (2) instead of IMC (with both $X, Y$ are set as $Z$), and then runs $k$-means on the top-$k$ eigenvectors of the completed matrix to obtain a clustering. This algorithm can be viewed as an improved version of MCCC to handle noisy features $Z$.

We now compare our algorithm with $k$-means, signed graph clustering with matrix completion [12] (SignMC) and MCCC [37]. Note that since MCCC has been shown to outperform most other state-of-the-art semi-supervised clustering algorithms in [37], comparing with MCCC is sufficient to demonstrate the effectiveness of our algorithm. We perform each method on three real-world datasets: Mushrooms, Segment and Covtype [1]. All of them are classification benchmarks where features and ground-truth class of items are both available, and their statistics are summarized in Table 2. For each dataset, we randomly sample $m = [1, 5, 10, 15, 20, 25, 30] \times n$ pairwise constraints, and perform each algorithm to derive a clustering $\pi$, where $\pi_i$ is the cluster index of item $i$. We then evaluate $\pi$ by the following pairwise error to ground-truth:

$$\frac{n(n-1)}{2}\bigg(\sum_{(i,j):\pi_i^*=\pi_j^*} \mathbf{1}(\pi_i \neq \pi_j) + \sum_{(i,j):\pi_i^*\neq\pi_j^*} \mathbf{1}(\pi_i = \pi_j)\bigg)$$

where $\pi_i^*$ is the ground-truth class of item $i$.

Figure 2 shows the result of each method on all three datasets. We first see that for Mushrooms dataset where features are perfect (100% training accuracy can be attained by linear-SVM for classification), both MCCC and DirtyIMC can obtain a perfect clustering, which shows that MCCC is indeed effective with perfect features. For Segment and Covtype datasets, we observe that the performance of $k$-means and MCCC are dominated by feature quality. Although MCCC still benefits from constraint information as it outperforms $k$-means, it clearly does not make the best use of constraints, as its performance does not improves even if number of constraints increases. On the other hand, the error rate of SignMC can always decrease down to 0 by increasing $m$. However, since it disregards features, it suffers from a much higher error rate than methods with features when constraints are few. We again see DirtyIMC combines advantage from MCCC and SignMC, as it makes use of features when few constraints are observed yet leverages constraint information simultaneously to avoid being trapped by feature noise. This experiment shows that our model outperforms state-of-the-art approaches for semi-supervised clustering.

**Acknowledgement.** We thank David Inouye and Hsiang-Fu Yu for helpful comments and discussions. This research was supported by NSF grants CCF-1320746 and CCF-1117055.

## Footnotes

[1] All datasets are available at `http://www.csie.ntu.edu.tw/~cjlin/libsvmtools/datasets/`. For Covtype, we subsample from the entire dataset to make each cluster has balanced size.

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
