[Supplementary Material]

**Algorithm 1** Alternative Minimization for DirtyIMC with Squared Loss
---
**Input:** feature matrix $X$, $Y$, parameters $(\lambda_M, \lambda_N)$ in objective (2), max iteration $t_{max}$.
$t = 0, M^{(t)} \leftarrow 0, N^{(t)} \leftarrow 0$.
**while** Not converged and $t < t_{max}$ **do**
    Solve $M^{(t+1)} \leftarrow \arg\min_M \sum_{(i,j)\in\Omega} (XMY_{ij}^T - (R - N^{(t)})_{ij})^2 + \lambda_M \|M\|_*$
    Solve $N^{(t+1)} \leftarrow \arg\min_N \sum_{(i,j)\in\Omega} (N_{ij} - (R - XM^{(t+1)}Y^T)_{ij})^2 + \lambda_N \|N\|_*$
    $t \leftarrow t + 1$
**end while**
**return** recovered matrix $XM^{(t)}Y^T + N^{(t)}$.
---

## Appendix A: Solving DirtyIMC Objectives

To solve problem (2), we propose an alternative minimization scheme where at each step we fix one of the variables ($M$ or $N$) and solve for the other. For simplicity, here we focus on the case where $\ell$ is squared loss, which is also considered in our experiments. The algorithm is summarized in Algorithm 1. As one variable is fixed, the subproblem reduces to either standard matrix completion or IMC, which is easy to solve as discussed below. This algorithm can be viewed as applying a block coordinate descent algorithm on convex (but non-smooth) function, and thus is guaranteed to converge to global optimal using standard analysis (e.g. [15]).

We now briefly discuss how to solve two subproblems in Algorithm 1. First, when fixing $N$, the subproblem becomes an IMC objective with observed matrix to be $R - N$. We then apply proximal gradient descent to update $M$. Notice that in our setting, feature dimensions $(d_1, d_2)$ are much smaller than number of entities $(n_1, n_2)$. Therefore, for small $d$, it is relatively inexpensive to compute a full SVD for a $d_1 \times d_2$ matrix in each proximal step.

On the other hand, when fixing $M$, the subproblem becomes standard matrix completion problem for the residual matrix $R - XMY^T$. We then apply active subspace selection algorithm (Active-ALT) [17] to solve the matrix completion problem.

Another possibility is to consider the non-convex relaxation of problem (2) as:

$$\min_{U,V,W,H} \sum_{(i,j)\in\Omega} \ell((XU^TVY^T + W^TH)_{ij}, R_{ij}) + \frac{\lambda_M}{2}(\|W\|_F^2 + \|H\|_F^2) + \frac{\lambda_N}{2}(\|U\|_F^2 + \|V\|_F^2), \quad (5)$$

in which $M$, $N$ is factorized to low rank matrices $U \in \mathbb{R}^{d_1 \times k_1}, V \in \mathbb{R}^{d_2 \times k_1}$ and $W \in \mathbb{R}^{n_1 \times k_2}, H \in \mathbb{R}^{n_2 \times k_2}$. A similar alternative minimization scheme, i.e. fix three variables and solve for the other, can be applied to obtain a solution for $U, V, W, H$. Although problem (5) is equivalent to the convex problem (2) if $k_1 \geq \text{rank}(M^*)$ and $k_2 \geq \text{rank}(N^*)$ [34], it is not jointly convex for all variables. So unlike Algorithm 1, using alternative minimization to solve (5) may not obtain the global optimum. However, the analysis in [3] shows that the algorithm converges to stationary points if each subproblem has a unique minimizer, which is indeed the case in (5) because of the regularizations. Researchers found that such non-convex relaxation to be useful since it is easier to solve, and empirically yields a competitive result compared to convex problem [22].

Finally, we notice that a recently proposed method "Boosted IMC" [33] could also be represented as a special case of our alternative scheme for non-convex relaxation (5). The method could be viewed as an one iteration heuristic of Algorithm 1 (i.e. $t_{max} = 1$), in which they first solve $N^{(1)}$ and then solve $M^{(1)}$ using matrix factorization. Although this method is proposed as a heuristic for Blog recommendation rather than an algorithm for solving a formal defined matrix completion objective, it could also be interpreted as an algorithm that approximately solves our DirtyIMC model. We also compare our DirtyIMC with Boosted IMC in Appendix C.

## Appendix B: Proofs

### Proof of Lemma 2

*Proof.* To begin with, we introduce a lemma to bound the Rademacher complexity for the function class with bounded trace norm.

**Lemma 4.** *Let* $S_w = \{W \in \mathbb{R}^{n \times n} \mid \|W\|_* \leq \mathcal{W}\}$ *and* $\mathcal{A} = \max_i \|A_i\|_2$, *where each* $A_i \in \mathbb{R}^{n \times n}$, *then:*

$$\mathbb{E}_\sigma \Big[ \sup_{W \in S_w} \frac{1}{m} \sum_{i=1}^m \sigma_i trace(W A_i) \Big] \leq 2\mathcal{A}\mathcal{W}\sqrt{\frac{\log 2n}{m}}.$$

This Lemma is a special case of Theorem 1 in [20] with the fact that the dual norm of the matrix 2-norm is trace norm. Thus, by using Rademacher contraction principle (e.g. Lemma 5 in [27]), $\Re(F_\Theta)$ can be written as:

$$\Re(F_\Theta) \leq L_\ell \mathbb{E}_\sigma \Big[ \sup_{\theta \in \Theta} \frac{1}{m} \sum_{\sigma=1}^m \sigma_\alpha (XMY^T + N)_{i_\alpha j_\alpha} \Big]$$

$$= L_\ell \mathbb{E}_\sigma \Big[ \sup_{\|M\|_* \leq \mathcal{M}} \frac{1}{m} \sum_{\sigma=1}^m \sigma_\alpha \mathbf{x}_{i_\alpha}^T M \mathbf{y}_{j_\alpha} \Big] + L_\ell \mathbb{E}_\sigma \Big[ \sup_{\|N\|_* \leq \mathcal{N}} \frac{1}{m} \sum_{\sigma=1}^m \sigma_\alpha N_{i_\alpha j_\alpha} \Big]$$

$$= L_\ell \mathbb{E}_\sigma \Big[ \sup_{\|M\|_* \leq \mathcal{M}} \frac{1}{m} \sum_{\alpha=1}^m \sigma_\alpha \mathrm{trace}(M \mathbf{y}_{j_\alpha} \mathbf{x}_{i_\alpha}^T) \Big] + L_\ell \mathbb{E}_\sigma \Big[ \sup_{\|N\|_* \leq \mathcal{N}} \frac{1}{m} \sum_{\alpha=1}^m \sigma_\alpha \mathrm{trace}(N \mathbf{e}_{j_\alpha} \mathbf{e}_{i_\alpha}^T) \Big]$$

$$\leq 2L_\ell \Big( \mathcal{M} \max_{i,j} \|\mathbf{y}_j \mathbf{x}_i^T\|_2 \sqrt{\frac{\log 2d}{m}} + \mathcal{N} \sqrt{\frac{\log 2n}{m}} \Big),$$

where the last equation is derived by applied Lemma 4. Since $\max_{i,j} \|\mathbf{y}_j \mathbf{x}_i^T\|_2 = \max_j \|\mathbf{y}_j\|_2 \max_i \|\mathbf{x}_i\|_2$, we derive an upper bound of $\Re(F_\Theta)$:

$$\mathbb{E}_\Omega \big[ \Re(F_\Theta) \big] \leq 2L_\ell \mathcal{M} \mathcal{X} \mathcal{Y} \sqrt{\frac{\log 2d}{m}} + 2L_\ell \mathcal{N} \sqrt{\frac{\log 2n}{m}}. \tag{6}$$

However, in some circumstances, the above bound (6) will become too loose for our sample complexity analysis. As a result, we need to deal with these cases by introducing a tighter bound on trace norm of residual (i.e. $\mathcal{N}$). The following bound mainly follows the proof step in [32], which provides a tighter bound on trace-norm regularized function class. To begin with, we can rewrite $\Re(F_\Theta)$ as:

$$\Re(F_\Theta) = \mathbb{E}_\sigma \Big[ \sup_{f \in F_\Theta} \frac{1}{m} \sum_{\alpha=1}^m \sigma_\alpha \ell(f(i_\alpha, j_\alpha), R_{i_\alpha, j_\alpha})) \Big]$$

$$= \mathbb{E}_\sigma \Big[ \sup_{f \in F_\Theta} \frac{1}{m} \sum_{(i,j)} \Gamma_{ij} \ell(f(i,j), R_{ij}) \Big],$$

where $\Gamma \in \mathbb{R}^{n_1 \times n_2}$ with each entry $\Gamma_{ij} = \sum_{\alpha : i_\alpha = i, j_\alpha = j} \sigma_\alpha$. Now, using the same trick in [32], we can divide $\Gamma$ based on the "hit-time" on entry $(i, j)$ of $\Omega$, with some threshold $p > 0$ whose value will be set later. Formally, let $h_{ij} = |\{\alpha : i_\alpha = i, j_\alpha = j\}|$, and let $A, B \in \mathbb{R}^{n_1 \times n_2}$ be defined as:

$$A_{ij} = \begin{cases} \Gamma_{ij}, & \text{if } h_{ij} > p, \\ 0, & \text{otherwise.} \end{cases} \qquad B_{ij} = \begin{cases} 0, & \text{if } h_{ij} > p, \\ \Gamma_{ij}, & \text{otherwise.} \end{cases} \tag{7}$$

By construction, $\Gamma = A + B$. Therefore, we can separate $\Re(F_\Theta)$ as:

$$\Re(F_\Theta) = \mathbb{E}_\sigma \Big[ \sup_{f \in F_\Theta} \frac{1}{m} \sum_{(i,j)} A_{ij} \ell(f(i,j), R_{ij}) \Big] + \mathbb{E}_\sigma \Big[ \sup_{f \in F_\Theta} \frac{1}{m} \sum_{(i,j)} B_{ij} \ell(f(i,j), R_{ij}) \Big]. \tag{8}$$

For the first term of (8), by the assumption $|\ell(f(i,j), R_{ij})| \leq \mathcal{B}$, it can be upper bounded by:

$$\frac{\mathcal{B}}{m} \mathbb{E}_\sigma \Big[ \sum_{(i,j)} |A_{ij}| \Big] \leq \frac{\mathcal{B}}{\sqrt{p}}$$

by using the Lemma 10 in [32]. Now consider the second term of (8). Again, by using Rademacher contraction principle, it can be upper bounded by:

$$\frac{L_\ell}{m} \mathbb{E}_\sigma \Big[ \sup_{f \in F_\Theta} \sum_{(i,j)} B_{ij} f(i,j) \Big]$$

$$= \frac{L_\ell}{m} \mathbb{E}_\sigma \Big[ \sup_{M : \|M\|_* \leq \mathcal{M}} \sum_{(i,j)} B_{ij} \mathbf{x}_i^T M \mathbf{y}_j \Big] + \frac{L_\ell}{m} \mathbb{E}_\sigma \Big[ \sup_{N : \|N\|_* \leq \mathcal{N}} \sum_{(i,j)} B_{ij} N_{ij} \Big], \tag{9}$$

which is separated by feature-covered part and residual part. We first consider the residual part (i.e. the second term of (9)). By applying Hölder's inequality, the second term of (9) is upper bounded by:

$$\frac{L_\ell}{m}\sup_{N:\|N\|_*\leq\mathcal{N}}\|B\|_2\|N\|_* = \frac{L_\ell\mathcal{N}}{m}\mathbb{E}_\sigma\big[\|B\|_2\big] \leq \frac{2.2CL_\ell\mathcal{N}\sqrt{p}(\sqrt{n_1}+\sqrt{n_2})}{m},$$

where the last inequality is derived by applying Lemma 11 in [32]. Now, for the first term of (9), notice that we can upper bound this term by:

$$\frac{L_\ell}{m}\mathbb{E}_\sigma\Big[\sup_{M:\|M\|_*\leq\mathcal{M}}\sum_{\alpha=1}^m\sigma_\alpha\mathbf{x}_{i_\alpha}^T M\mathbf{y}_{j_\alpha}\Big]$$

$$= L_\ell\mathbb{E}_\sigma\Big[\sup_{\|M\|_*\leq\mathcal{M}}\frac{1}{m}\sum_{\alpha=1}^m\sigma_\alpha\mathrm{trace}(M\mathbf{y}_{j_\alpha}\mathbf{x}_{i_\alpha}^T)\Big]$$

$$\leq 2L_\ell\mathcal{M}\max_{i,j}\|\mathbf{y}_j\mathbf{x}_i^T\|_2\sqrt{\frac{\log 2d}{m}}$$

$$= 2L_\ell\mathcal{M}\mathcal{X}\mathcal{Y}\sqrt{\frac{\log 2d}{m}}.$$

Therefore, putting back all above upper bound to (8), with $p$ chosen to be $m\mathcal{B}/(2.2CL_\ell\mathcal{N}(\sqrt{n_1}+\sqrt{n_2}))$, we can get another bound on $\mathfrak{R}(F_\Theta)$ by:

$$\mathbb{E}_\Omega\big[\mathfrak{R}(F_\Theta)\big] \leq 2L_\ell\mathcal{M}\mathcal{X}\mathcal{Y}\sqrt{\frac{\log 2d}{m}} + \sqrt{9CL_\ell\mathcal{B}\frac{\mathcal{N}(\sqrt{n_1}+\sqrt{n_2})}{m}}. \tag{10}$$

The Theorem thus follows by combining two bounds from (6) and (10). $\qquad\square$

**Proof of Lemma 3**

We first need the following lemma to bound the largest singular value $\sigma_x$ of feature matrix $X$ (and also $\sigma_y$ of $Y$).

**Lemma 5.** *Let $X \in \mathbb{R}^{n\times d}$ be a feature matrix. Then there exists a constant $C''$ (i.e. not a function of $n$), such that:*

$$\sigma_x \geq C''\gamma\mathcal{X}\sqrt{n}.$$

*Proof.* Let $\tilde{\mathbf{x}}_i$ be normalized feature vectors that $\tilde{\mathbf{x}}_i = \frac{\mathbf{x}_i}{\|\mathbf{x}_i\|}$ for all $i = 1\ldots n$, so that each $\tilde{\mathbf{x}}_i$ lies on the $d$ dimensional unit sphere $S_d = \{\tilde{\mathbf{x}} \in \mathbb{R}^d \mid \|\tilde{\mathbf{x}}\| = 1\}$. From Lemma 21 of [14], for any $\eta > 0$, the $d$ dimensional unit sphere can be partitioned into $N = (c/\eta)^d$ equal volume cells (denoted as $P_1\ldots P_N$) whose diameter is at most $\eta$, where $c$ is some constant. Therefore, if two unit vectors $\mathbf{x}, \mathbf{y}$ are in the same cell $P_i$, since $\|\mathbf{x} - \mathbf{y}\| \leq \eta$, the angle $\theta$ between $\mathbf{x}$ and $\mathbf{y}$ will satisfy

$$\frac{\theta}{2} \leq \sin^{-1}(\frac{\eta}{2\|\mathbf{x}\|}) = \sin^{-1}\frac{\eta}{2},$$

which leads the inner product of $\mathbf{x}$ and $\mathbf{y}$ to be:

$$\mathbf{x}^T\mathbf{y} = \cos\theta = 1 - 2\sin^2(\frac{\theta}{2}) \geq 1 - 2(\frac{\eta}{2})^2 = 1 - \frac{\eta^2}{2}.$$

Thus, taking $\eta = 1$, we can partition the unit sphere into $N = c^d$ cells such that

$$\mathbf{x}^T\mathbf{y} \geq \frac{1}{2}, \text{ if } \mathbf{x}, \mathbf{y} \in P_i.$$

Now reconsider $n$ normalized feature vectors $\tilde{\mathbf{x}}_1, \ldots, \tilde{\mathbf{x}}_n$, each of which belongs to one of the cell $P_i$. By Pigeonhole Theorem, there exists one cell $P^* \in \{P_i\}_{i=1}^N$ such that at least $n/N$ vectors lie

in $P^*$. Consider any unit vector $\mathbf{w}$ in $P^*$, then we have $\tilde{\mathbf{x}}_i^T \mathbf{w} \geq \frac{1}{2}$ for all $\tilde{\mathbf{x}}_i \in P^*$. Therefore,

$$
\begin{aligned}
\|X\mathbf{w}\|_2 &\geq \sqrt{\sum_{i:\mathbf{x}_i \in P^*} (\mathbf{x}_i^T \mathbf{w})^2} \\
&\geq \sqrt{\sum_{i:\mathbf{x}_i \in P^*} \gamma^2 \mathcal{X}^2 (\tilde{\mathbf{x}}_i^T \mathbf{w})^2} \\
&\geq \gamma \mathcal{X} \sqrt{\frac{n}{N}(\frac{1}{2})^2} \\
&= \left(\frac{1}{2\sqrt{N}}\right) \gamma \mathcal{X} \sqrt{n},
\end{aligned}
$$

which concludes that

$$
\sigma_x \geq C'' \gamma \mathcal{X} \sqrt{n}
$$

where $C'' = \frac{1}{2\sqrt{N}}$ is a constant with respect to $n$. $\qquad\square$

With Lemma 5, we can now prove the Lemma 3 as follows:

*Proof.* To begin with, we have:

$$
\|X_\mu^T R Y_\nu\|_2 \leq \|X_\mu\|_2 \|R\|_2 \|Y_\nu\|_2 \leq \sigma_x \sigma_y \|R\|_*.
$$

On the other hand, by the closed form solution of $\hat{M}$, we have:

$$
\begin{aligned}
\|\hat{M}\|_* &\leq \|\hat{M}\|_2 \hat{d} \\
&= \|(X_\mu^T X_\mu)^{-1} X_\mu^T R Y (Y_\nu^T Y_\nu)^{-1}\|_2 \hat{d} \\
&\leq \frac{\sigma_x \sigma_y \|R\|_* \hat{d}}{\sigma_{xm}^2 \sigma_{ym}^2},
\end{aligned}
$$

where $\sigma_{xm}$, $\sigma_{ym}$ are the smallest singular value of $X_\mu$, $Y_\nu$ respectively. Also, by construction of $X_\mu$ and $Y_\nu$, we have $\sigma_{xm} \geq \mu\sigma_x$ and $\sigma_{ym} \geq \nu\sigma_y$. Combining Lemma 5, we have:

$$
\begin{aligned}
\|\hat{M}\|_* &\leq \frac{\|R\|_* \hat{d}}{\mu^2 \nu^2 \sigma_x \sigma_y} \\
&\leq \frac{\|R\|_* \hat{d}}{C' \sqrt{n_1 n_2} \gamma^2 \mu^2 \nu^2 \mathcal{X}\mathcal{Y}},
\end{aligned}
$$

where $C'$ is a constant independent to $n_1, n_2$. By the fact that $\|R\|_* \leq \mathcal{R}\sqrt{n_1 n_2}$, the lemma is proved. $\qquad\square$

**Proof of Theorem 2**

*Proof.* By the construction of feature space, we can rewrite $X$ and $Y$ as follows:

$$
X = \sum_{i=1}^{t-1} \mathbf{u}_i \mathbf{e}_i^T + \sum_{i=t}^{d} \tilde{\mathbf{u}}_i \mathbf{e}_i^T \qquad Y = \sum_{i=1}^{t-1} \mathbf{v}_i \mathbf{e}_i^T + \sum_{i=t}^{d} \tilde{\mathbf{v}}_i \mathbf{e}_i^T, \tag{11}
$$

where for each $\tilde{\mathbf{u}}_i$, $\tilde{\mathbf{u}}_i^T \mathbf{u}_j = 0, \forall j$. Therefore, the trace norm of residual can be bounded by:

$$
\begin{aligned}
\|R - X\hat{M}Y^T\|_* &= \|\tilde{U}\tilde{U}^T R + R\tilde{V}\tilde{V}^T - \tilde{U}\tilde{U}^T R\tilde{V}\tilde{V}^T\|_* \\
&\leq 2\|\tilde{U}\tilde{U}^T U\Sigma V^T\|_* + \|U\Sigma V^T \tilde{V}\tilde{V}^T\|_* \\
&\leq 3\sum_{i=t}^{k} \sigma_i,
\end{aligned}
$$

where $\tilde{U}, \tilde{V}$ are the second term of $X$ and $Y$ in (11). Moreover, we have $\sigma_i = o(\sqrt{n})$ for all $i \geq t$. To see this, suppose $\sigma_p = \Omega(\sqrt{n})$ for any $t \leq p \leq k$, then:

$$\lim_{n \to \infty} \frac{\sigma_t}{\sqrt{n}} \geq \lim_{n \to \infty} \frac{\sigma_p}{\sqrt{n}} > 0,$$

leading a contradiction to the definition of $\sigma_t$. Therefore we can conclude:

$$\mathcal{N} = \|R - X\hat{M}Y^T\|_* \leq 3 \sum_{i=t}^{k} \sigma_i \leq 3k \times o(\sqrt{n}) = o(\sqrt{n}),$$

and the Theorem is thus proved by plugging the above bound to Corollary 1. $\qquad\square$

**Proof of Theorem 3**

*Proof.* We prove the Theorem by showing that the trace norm of $R - X\hat{M}Y^T$ will be $O((g(n) + h(n)) \log n)$ in this scenario given that other dimensions ($d$ and $k$) do not grow as a function of $n$. First, note that in this scenario, we can denote $X = U + \Delta U$ and $Y = V + \Delta V$, where $U \subseteq \mathrm{col}(R), V \subseteq \mathrm{row}(R)$ and $\Delta U, \Delta V$ are $g(n), h(n)$ column sparse respectively. The following Lemma then bounds the trace norm of $R - X\hat{M}Y^T$ in terms of $\Delta U$ and $\Delta V$.

**Lemma 6.** *Let $\Delta U, \Delta V$ be defined as above. Then with high probability,*

$$\|R - X\hat{M}Y^T\|_* \leq c_1 \xi_1 \sqrt{\frac{k}{g(n)}} \|\Delta U^T R\|_* + c_2 \xi_2 \sqrt{\frac{k}{h(n)}} \|R \Delta V\|_* \tag{12}$$

*with some universal constants $c_1$ and $c_2$.*

*Proof.* Let $\Delta U = U_1 \Sigma_1 V_1^T$ and $\Delta V = U_2 \Sigma_2 V_2^T$ be the reduced SVD of the perturbation matrix $\Delta U, \Delta V$ accordingly. Then we have:

$$\begin{aligned}
\|R - X\hat{M}Y^T\|_* &\leq \|U_1 U_1^T R + R U_2 U_2^T - U_1 U_1^T R U_2 U_2^T\|_* \\
&\leq 2\|U_1 U_1^T R\|_* + \|R U_2 U_2^T\|_* \\
&= 2\|\Delta U(V_1 \Sigma_1^{-2} V_1^T)\Delta U^T R\|_* + \|R\Delta V(V_2 \Sigma_2^{-2} V_2^T)\Delta V^T\|_*. \tag{13}
\end{aligned}$$

For the first term of (13), using Hölder's inequality, we can upper bound it by:

$$\|\Delta U\|_2 \|V_1 \Sigma_1^{-2} V_1^T\|_2 \|\Delta U^T R\|_* = \|\Delta U\|_2 \|\Sigma_1^{-2}\|_2 \|\Delta U^T R\|_*, \tag{14}$$

which suggests that we need to bound the largest and smallest singular values of $\Delta U$ to bound (14). Consider $\Delta U' \in \mathbb{R}^{g(n) \times k}$ to be the truncated $\Delta U$ where only non-zero rows in $\Delta U$ are left. The spectrum of $\Delta U'$ is same as $\Delta U$. Moreover, its two norm can be bounded by:

$$\|\Delta U'\|_2 \leq \|\xi_1 E_1\|_2 \leq \xi_1 \sqrt{kg(n)},$$

where $E_1 \in \mathbb{R}^{g(n) \times k}$ is the matrix with all entries are one. Also, using the result of [31], we can guarantee that with high probability $\sigma_k(\Delta U') \geq \Omega(\sqrt{g(n)} - \sqrt{k})$, which suggests w.h.p.:

$$\|\Sigma_1^{-2}\| = \frac{1}{\sigma_k(\Delta U)^2} = \frac{1}{\sigma_k(\Delta U')^2} \leq O(\frac{1}{g(n)}).$$

Thus, combining the above two bounds, the first term of (13) can be upper bounded by:

$$c_1 \xi_1 \sqrt{\frac{k}{g(n)}} \|\Delta U^T R\|_*,$$

with some universal constant $c_1$. Similarly, the second term of (13) can be upper bounded by $c_2 \xi_2 \sqrt{k/h(n)} \|R \Delta V\|_*$. The lemma is thus proved. $\qquad\square$

Therefore, given Lemma 6, we now need to bound $\|\Delta U^T R\|_*$ and $\|R\Delta V\|_*$. We first focus on bounding the term $\|\Delta U^T R\|_*$. By $R = UV^T$ and the construction of $U, V$, we have:

$$\|\Delta U^T R\|_* = \|\Delta U^T UV^T\|_* \leq \|GV^T\|_*$$

where $G \in \mathbb{R}^{k \times k}$ with each entry in $G_{ij} \sim \xi_1 g(n)\mathcal{N}(0, \sigma^2)$. Thus, let $Z = GV^T$, $Z \in \mathbb{R}^{k \times n}$, then each entry $Z_{ij} \sim \xi_1 g(n)\frac{\sigma^2}{2}\chi_k^2$, where $\chi_k^2$ is a chi-square distribution with degree of freedom $k$.

We next show that the trace norm of $Z$ will be bounded in small enough order with high probability. To begin with, the following Lemma is used as an exponentially decreasing bound on the tail distribution of chi-square statistics.

**Lemma 7** (Exponential Tail Bound of $\chi_k^2$). *Let $X$ be a random variable which follows $\chi_k^2$. Then for any $t > 1$, we have:*

$$\Pr(X \geq tk) \leq \exp\left\{\frac{-k(\sqrt{(t-1)^2 + 1} - 1)}{2}\right\}$$

This Lemma is a corollary of Lemma 1 in [23]. Given this lemma, we can now derive the following lemma to upper bound $\|\Delta U^T R\|_*$:

**Lemma 8.** *Let $\Delta U^T R \in \mathbb{R}^{k \times n}$ where $\Delta U$ and $R$ are set as in Theorem 3. Then its trace norm can be upper bounded by:*

$$\|\Delta U^T R\|_* \leq C_1 k^{\frac{3}{2}} g(n)\sqrt{n}\log n$$

*with probability at least $1 - kn^{-\frac{k-2}{2}}$.*

*Proof.* Since $\|\Delta U^T R\|_* \leq \|Z\|_*$ where $Z_{ij} \sim \xi_1 g(n)\frac{\sigma^2}{2}\chi_k^2$, by applying Lemma 7 with $t = \log n$, we can guarantee that with probability at least $1 - n^{\frac{-k}{2}}$:

$$Z_{ij} \leq \xi_1 g(n)\frac{\sigma^2}{2}k\log n.$$

Thus, by applying union bound on each $Z_{ij}$, with probability at least $1 - kn^{-\frac{k-2}{2}}$:

$$\|\Delta U^T R\|_* \leq \|Z\|_* \leq \xi_1 g(n)\frac{\sigma^2}{2}k\log n\|E\|_*,$$

where $E \in \mathbb{R}^{k \times n}$ is a rank-1 matrix with all entries are 1. We can thus conclude the Lemma by the fact that $\|E\|_* = \|E\|_2 = \sqrt{nk}$. $\square$

Similarly, by using the same proof steps, it could also be shown that $\|R\Delta V\|_* \leq C_2 k^{3/2} h(n)\sqrt{n}\log n$. Therefore, substituting above bounds back to Lemma 6, we obtain:

$$\begin{aligned}
\mathcal{N} &= \|R - X\hat{M}Y^T\|_* \\
&\leq c_1\xi_1\sqrt{\frac{k}{g(n)}}\|\Delta U^T R\|_* + c_2\xi_2\sqrt{\frac{k}{h(n)}}\|R\Delta V\|_* \\
&= O\left(\max(\sqrt{g(n)}, \sqrt{h(n)})\sqrt{n}\log n\right),
\end{aligned}$$

and the proof is thus completed by plugging this result into Corollary 1. $\square$

## Appendix C: More Synthetic experiments for DirtyIMC

### Experiment on random orthogonal model

Here we conduct an experiment based on random orthogonal model stated in Theorem 2. We create a low rank matrix $R = U\Sigma V^T$ where $U, V \in \mathbb{R}^{n \times 20}$ are both random orthogonal matrix, and the singular values to be $\sqcup_{\alpha=1}^{10}\{\alpha n, \alpha\log n\}$, so there are 10 singular values have smaller growth rate $O(\log n)$. Follow Theorem 2, we construct $X, Y$ by replacing the bottom 10 singular vectors in

Figure 3: A synthetic experiment where noise only corrupts the insignificant part of true latent features (i.e. space spanned by smaller singular values). We see that in this case, given $O(n)$ observations, DirtyIMC could still recover the underlying matrix using sufficiently informative features, while matrix completion fails to recover as the error becomes unbounded with larger $n$. The result supports guarantee provided in Theorem 2.

$U$ and $V$ with bases orthogonal to $U$ and $V$. We increase $n$ from 250 to 3000, and for each $n$ we randomly sample $m = 100n$ observations, apply our model and matrix completion to complete the matrix, and evaluate the recovered matrix using relative error. From Theorem 2, our DirtyIMC model should be able to approximately recover the matrix given $100n > o(n)$ observations, which is indeed true as Figure 3 suggests. As a comparison, standard matrix completion fails to recover the matrix with only $O(n)$ observations as $n$ increases. This result empirically supports our theoretical analysis on the usefulness of noisy features.

**Finer results for synthetic experiments in Section 5**

Figure 4 and 5 show finer plots under each sparsity of observation $\rho_s$ and feature noise level $\rho_f$.

**Comparisons between DirtyIMC and Boosted IMC**

As we mentioned in Appendix A, a recently proposed method "Boosted IMC" [33] could be viewed as a special case of our model, where their method is basically Algorithm 1 with $t_{max} = 1$, and in each subproblem they replace the trace norm regularized objective with matrix factorization objective. Here we compare our DirtyIMC (Algorithm 1) with Boosted IMC on synthetic datasets generated as same as Section 5 stated. We follow their implementation with rank of $U, V, W, H$ are all set to be 40. The result is shown in Figure 6.

We observe that though Boosted IMC has a similar trend to DirtyIMC, in general, DirtyIMC performs better than Boosted IMC. However, Boosted IMC may be still good enough as an approximation of DirtyIMC in certain cases where efficiency is critical, since it only requires one iteration update of $M$ and $N$.

## Appendix D: Details for applying DirtyIMC to semi-supervised clustering

Here we follow the discussion in Section 5 for semi-supervised clustering. Suppose we are given $m$ pairwise constraints describing similarity (or dissimilarity) of some pairs of items, then we can construct the following pairwise similarity matrix $S$ as:

$$S_{ij} = \begin{cases} 1, & \text{if } i \text{ and } j \text{ are similar}, \\ 0, & \text{if } i \text{ and } j \text{ are dissimilar}. \end{cases}$$

Obviously, $S$ has many missing entries since only $m \ll n^2$ pairwise constraints are known. In addition, ideally $S$ should be a subset of observations sampled from $UU^T$, where $U \in \mathbb{R}^{n \times k}$ with each $i$-th column of $U$ is an indicator vector of the $i$-th cluster. Therefore, one can try to recover

Figure 4: Finer results for synthetic experiments where completion methods are applied under different feature quality with a fixed $\rho_s$

(or complete) the matrix back with DirtyIMC objective, and the column space of recovered matrix, spanned by its top-$k$ eigenvectors, will (ideally) reveal the indicator vectors. Our detailed algorithm is summarized in Algorithm 2.

One subtle yet critical issue in Algorithm 2 is to compute the top-$k$ eigenvectors of recovered $S$ (denoted as $R$). Note that after solving DirtyIMC objective, we are only given the low rank expression of $N^*$ and $M^*$. Compute $R$ explicitly and then compute its leading eigenvectors is expensive and not scalable. Therefore, we instead run subspace iteration on $N^* + Z\tilde{M}^*Z^T$ to solve for top-$k$ eigenvectors efficiently. Also, since the resulting top-$k$ eigenvectors are used for running $k$-means, we do not need to obtain a very accurate eigenvectors in this case. Therefore, parameters associated with precision ($t_{max}$ and $\epsilon$) could be set relatively loose for efficiency in practice.

Figure 5: Finer results for synthetic experiments where completion methods are applied under different sparsity of observations with a fixed $\rho_f$

Figure 6: Performance of DirtyIMC and Boosted IMC (an approximation of DirtyIMC model) on synthetic datasets.

---

**Algorithm 2** Semi-supervised clustering with DirtyIMC

**Input:** feature matrix $Z$, pairwise similarity matrix $S$, number of clusters $k$, regularization parameters $(\lambda_M, \lambda_N)$ in (2).
// Solve DirtyIMC objective with Algorithm 1.
$(M^*, N^*) \leftarrow \arg\min_{M,N} \sum_{(i,j)\in S}((ZMZ^T + N)_{ij} - S_{ij})^2 + \lambda_M\|M\|_* + \lambda_N\|N\|_*$
// Subspace iterations for finding top-$k$ eigenvectors.
$\epsilon \leftarrow 10^{-3}, t_{max} \leftarrow 10, t \leftarrow 1$
$[U_M, \Sigma_M, V_M] \leftarrow \text{SVD}(M^*)$
**initialize** $U_{(t)} \leftarrow \text{QR}(ZU_M, k)$
**while** $t \leq t_{max}$ **do**
$\quad U_{(t+1)} \leftarrow \text{QR}(ZM^*Z^TU_{(t)} + N^*U_{(t)}, k)$
$\quad t \leftarrow t + 1$
$\quad$ **if** $\sigma_k(U_{(t)}^T U_{(t+1)}) < \epsilon$ **then**
$\quad\quad$ **break**
$\quad$ **end if**
**end while**
$\text{idx} \leftarrow \text{kmeans}(U_{(t)}, k)$
**return** idx

---