[Reviews · NeurIPS 2015]

Submitted by Assigned_Reviewer_1

This work extends Xu et al. (2013) which leverages noiseless side information for matrix completion to the regime of noisy side information. The key idea is to decompose the low-rank structure into two components, each of which are regularized to be low-rank themselves: one communicated by the features (side information), and one from the actual noisy observations of the matrix entries. A theroetical bound is derived on the recovery accuracy that depends on a quantification of feature quality. Interestingly, this coincides with the O(n^1.5) of Xu et al (2013) complexity bound in the worst case where features are completely noisy, but also shows that we can do strictly better than this when features carry information.

One thing that is not provided here is guidance on how to choose the regularization weights $\lambda$, $\lambda_2$. It is mentioned at the end of Section 3 that these will be formally answered in Section 4, but it is unclear from my perspective how to translate the theoretical analysis into actual values for these regularization weights. In fact, in the experimental results, these regularization weights were chosen via brute-force search which may be intractable/subject to over-fitting in more realistic cases.

Nevertheless, the experimetnal results are compelling and effectively demonstrate the robustness of the proposed algorithm w.r.t. side information quality, as compared to other matrix completion algorithms that exploit side information.

Summary: This work contains significant

methodological and theoretical contributions to the problem of matrix completion with side information by extending previous work to the noisy feature, and connecting feature quality to theoretical bounds on recovery. Overall, the work is clearly explained, is of high quality, and the results effectively illustrate the advantage of the proposed method.

Submitted by Assigned_Reviewer_2

This paper is suitable for all level of readers.

Overall This paper is very well written.

The idea is novel while some assumptions may not be quite valid, For example, the authors assume the M matrix to be low rank, and did not prove that.

I understand R matrix is low rank, but why the smaller matrix M is also low rank?

I would expect more elaboration on this assumption.

The contribution and proposal is straight froward and the presentation is easy to follow.

I like the idea of updating the side information, and setting of the problem is nicely formulated.

The comments to help further improvement is the author did not discuss the details about the implementation.

Solving the optimization problem is not trivial, especially get to the global optimal.

I do believe there are some constraints or further assumptions needed before this problem is solvable.

The variation of solutions might be another problem.

how to find the global optimal.

the idea is quite original, especially the estimation of the transform matrix.

While the authors may need to read some of the related work on noisy side information.

Some may be quite inspiring as well in how to solve this type of problem.

And it is highly recommend the authors compare and discuss similar methods in literature and experimental section.

For example: Matrix Completion with Noise Noisy Matrix Completion Using Alternating Minimization Distance Metric Learning from Uncertain Side Information with Application to Automated Photo Tagging\

the problem itself is common and practically useful. the idea seems interesting but involves some assumption, which is my major concern can be some problem affecting the results.

The authors needs further investigate, either prove the assumptions are valid, or remove the assumptions and reformulate the problem in a more natural / organic way.

Summary: The paper proposed to use noisy side information to help matrix completion.

Instead of trust the perfect side information, the authors proposed to also optimize the side information in the objective function.

The idea is interesting but some technical details may needs further discussion and may be arguable.

Author Feedback
Author rebuttal: We thank all reviewers for comments. The following is the response to each reviewer.

To Reviewer_1:
The setting of lambda, lambda2 is implicitly stated in Thm 1, as a particular setting of {\mathcal M} and {\mathcal N} in (3) corresponds to a setting of lambda, lambda2 in (2). However, admittedly, since the setting of \mathcal M (and \mathcal N) are related to feature quality (i.e. how large is the projection of underlying matrix onto the feature space X, Y) and such information is unknown in practice, we need to choose lambda, lambda2 using cross-validation in realistic scenarios.

To Reviewer_2:
In our formulation, we break the target matrix into two parts, R = XMY^T + N. As noted, there are infinite solutions if we don't constrain on the solution space of M and N, as for any M we can let N = R-XMT^T. However, since R is low rank (says rank k), it is natural to seek a simple and explanatory solution where some of R's
subspace (says rank r) is spanned by feature part XMY^T and the remaining subspace (rank k-r) is spanned by N. And since XMY^T is low rank, it is reasonable to assume M is also low rank, i.e. only a small subspace of X and subspace of Y should be effective to choose to form a rank-r subspace. The same assumption (i.e. M is low rank) is commonly assumed in this line of research [18, 28, 34]. We would elaborate this part more on top of current explanation (l.123).

We introduce our optimization strategy in Appendix A (l.132, l.497), which alternatively solves M and N. Each subproblem becomes standard trace-norm regularized MC and IMC problems, and thus can be solved by some well-studied algorithms, such as proximal gradient descent. While we agree that the optimization is not trivial, it has relatively less contribution in the optimization part, as we borrow some well-studied solvers here. Thus we move the optimization details to Appendix due to space issue. In addition, for squared loss, we do not need any further assumptions to make problem (2) solvable. While we have pointed out the global optimal can be achieved (l.131, l.503), it is indeed possible that there are multiple global optimal since the problem is not strongly convex. However, our analysis is shown on any global solution of (3), and therefore, variation of solutions is not a problem in our case, as for sufficient samples and appropriate constraint setting,
any global solution will achieve small enough error as Thm 1 stated (given assumptions hold).

Thanks for pointing out some valuable related work. The first two works do not consider any feature and instead consider the noise that occurs in observations. The third work is more application oriented using metric learning. Although we also demonstrate our model on a similar application - semi-supervised clustering, our work aims to provide a more general treatment to noisy features on matrix completion. In addition, their "uncertain side information" in fact corresponds to similar/dissimilar information between items in semi-supervised clustering, which means the uncertainty they consider is also on observations, while the noise we consider is on features. We are happy to include these related work in our final submission.

To Reviewer_3:
Features are indeed useful for cold start problem in recommender systems. We will discuss these in related work. However, solving cold-start problem is not the main focus in this paper, as we aim to focus on the recoverability instead.

To Reviewer_5:
We will thoroughly check and fix grammatical errors in the final submission.

To Reviewer_6:
Our formulation is indeed intuitive yet significantly improves the performance of IMC in both synthetic and real applications when features are noisy. Regarding to the theoretical result of sample complexity, most of known results are either for the case without features [5, 7, 31], or only applicable given perfect features (i.e. features satisfy (1)) [34]. Our analysis is the first guarantee for general features.